# Deterioration Effects on Bricks Masonry in the Venice Lagoon Cultural Heritage: Study of the Main Façade of the Santa Maria dei Servi Church (14th Century)

Chiara Coletti [1,*], Ludovica Pia Cesareo [2], Jacopo Nava [1], Luigi Germinario [1], Lara Maritan [1], Matteo Massironi [1] and Claudio Mazzoli [1]

1  Department of Geosciences, Padova University, Via Gradenigo 6, 25131 Padova, Italy
2  Department of Molecular Sciences and Nanosystems, Ca' Foscari University, Via Torino 155, 30170 Venice, Italy
*  Correspondence: chiara.coletti@unipd.it

**Abstract:** Tidal exchange, capillary rise, water condensation-evaporation cycles, and crystallization of salts are the main causes of damage in historic brick buildings in Venice. The present study addressed these issues by proposing a study of twenty-three brick samples collected on the main façade of the Santa Maria dei Servi Church (14th century). The color, mineralogical composition, and texture of these samples were studied using standard methods such as spectrophotometry, X-ray powder diffraction (XRPD), optical microscopy (OM), and field emission scanning electron microscopy (FESEM). The presence of carbonates (calcite and dolomite) and newly formed silicate phases, such as gehlenite and diopside, provided indications of the temperatures reached during firing and suggested the absence of a good standardization in the production process. Meanwhile, XRPD and hyperspectral analysis (HA) detected sulfates (e.g., gypsum and mirabilite) as the main weathering products due to the salt decay process that affects monuments in the Venice lagoon environment. Moreover, secondary phases, such as Mg- and Ca-zeolites, occurred in bricks where the groundmass observed by OM was more vitrificated, and the XRPD patterns displayed the highest amorphous content. On-site mapping of sulfates and chlorophyll by HA was also performed on the main façade of the Church, highlighting the large presence of salts and biodeterioration.

**Keywords:** bricks; cultural heritage; decay; weathering process; damage; historical building; construction materials; firing process; salt crystallization; Venice lagoon

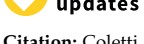

## 1. Introduction

### 1.1. "Venice and Its Lagoon"

The "Venice and its Lagoon" site was inscribed on the UNESCO World Heritage List in 1987 in recognition of its unique historical heritage and exceptional cultural traditions, integrated into an extraordinary natural landscape. The lagoon of Venice, indeed, is one of the most ancient and complex examples of the relationships between human activities and natural dynamics, hosting a high concentration of cultural patrimony accumulated over centuries. In this context, the combined action between natural weathering agents (rainfall, fog, snow, sea tides) and atmospheric pollutants (derived by the anthropogenic activity, strongly affected by the industrial output in the near city of Porto Marghera and the proximity of the Marco Polo Airport) [1] has a great impact on the cultural heritage conservation [2]. This is particularly evident in the exteriors of houses, palaces, and churches having bricks as the main building materials. The urban asset in Venice is, indeed, almost entirely built with brick, which has a weak structure that undergoes devastating effects with salts crystallization cycles [3] due to both the increasing sea variations of water level (daily tidal cycles and seasonal variations) [4–6] and sea spray, typical of the lagoon and sea environment.

The historical basements built using the Istria stone (a compact limestone) [7] are nowadays often immersed in salt water, and the bricks are subjected to water rise [8], causing differential wall decay. As storm tides called *acqua alta* increased in frequency (data available on Venice municipality website, https://www.comune.venezia.it accessed on 10 December 2022, Figure S1 in Supplementary Materials), the city's buildings were exposed to saltwater for several hours at a time. During water evaporation, salt growth can create pressure within the pore structure and induces defoliation, localized crumbling or spalling, powdering, and brick breakdown [9,10].

Many noteworthy buildings in Venice show strong losses of material at the bottom of the façades (usually in the 1–2 m from the soil level) due to the sub-florescence of salts which can induce scaling and cracking in bricks and subsequent loss of aesthetic quality of the surface and reduction of wall load-bearing capacity.

### 1.2. The Use of Bricks in Venice

Brick became the main material of construction used in Venice since the 12th century. The earliest Venice was built out of mud and wood for the construction of light buildings, suitable for lagoon soil but extremely vulnerable to fire. Many fires marked the history of the city of Venice (1106, 1120, 1149, 1167), which destroyed a large part of the old wooden city (including the Doge's Palace and the St Mark's Basilica) [11]. Thus, in order to face this issue, wooden masonries were converted to mixed structures, which, however, were gradually replaced by masonries mainly constructed with bricks.

The recycling of bricks and stones (mainly from the near Altino, Malamocco, and Aquileia), both for need and for symbolical reasons [12], was a practice in use until a local brick production started in Venetian furnaces. Meanwhile, in order to guarantee the reconstruction by bricks of wooden structures (public and private buildings), since the 12th century, bricks were produced locally, in particular where there were monumental construction sites, such as that in the Arsenal. Thus, after a long period of reuse of roman materials, Venetians began to use a local source, a compact clay named *caranto*, abundantly used in the first period of this new type of construction. The book *Capitolare de Fornaciis* (1368) attested to a large use of clay from nearby territories (such as Mestre) and later also documented clay materials from Treviso, Ferrara, and overall Padua as main sources.

During the 14th century, the Serenissima Republic of Venice was tempted to normalize the brick firing process in order to support local production and to increase the fired brick qualities in terms of standardization final aesthetical features and physical-mechanical properties. Although many authors addressed their research on mensiochronological aspects (relationship between brick size and their age) [12–14] and archaeometric analysis [15–17], nowadays, the knowledge of brick manufacture in Venice is still poor and lacking. Moreover, the practice of reuse of bricks can create chronological issues in terms of historical-architectural reconstruction and, therefore, also when addressing the most appropriate intervention in case of brick substitution. The present work aimed to fill part of these gaps by:

(i) a mineralogical and petrographic study of bricks of the Santa Maria dei Servi Church, since it did not have any interventions of restoration in its entire life (since the 14th century) and it was one of the first building sites constructed with the use of bricks from a local furnace; thus, results gave important information about bricks use and brick firing processes;

(ii) a decay mapping addressed to the main deterioration causes in the Venice monument: the detection of salts and biological growth by hyperspectral analysis, a detailed and non-destructive methodological approach to be applied in all the Venice urban assets;

(iii) the knowledge of the brick composition and the decay forms could pilot appropriate restorations for the studied site as well as for other cultural heritage in Venice.

### 1.3. The Santa Maria dei Servi Church

The Santa Maria dei Servi Church (a Servite Church), located in Sestiere Cannaregio (northern Venice), was one of the most important gothic buildings in Venice. The construction, which began in 1330 and was completed in 1491, is coeval to other gothic monumental churches such as the Basilica dei Santi Giovanni e Paolo and the Basilica di Santa Maria Gloriosa dei Frari (completed in 1430 and 1492, respectively). An approximated reconstruction of the Santa Maria dei Servi Basilica is possible through the analysis of maps and engravings. Pavon-Cauzzi [18] reported as results of a recent survey the following measurements 60 × 217 Venetian feet (20.86 × 75.45 m), while, Vicentini [19] reported 60 × 240 Venetian feet (20.86 × 83.4 m) using a report redacted in 1811, in which the apse was not considered [20]. Drawings by Jacopo De Barbari (1500) showed a prospective plant where the Santa Maria dei Servi Church showed the fifteen-century dome. In contrast, the main façade was reported by a drawing made by Luca Carlevarijs (1703), which stood out as the mono-cuspidate gothic façade made of bricks.

What remains of the Servite Church today is the main façade, the lateral portal, and the apse. The decline of the basilica began in 1769 when a fire destroyed part of the convent and the famous library. However, it was after the downfall of the Serenissima Republic of Venice (1797) and the subsequent arrival of Napoleonic troops (1806–1810) that Santa Maria dei Servi was slowly torn down. Altars were dismantled or sold, and all the paintings and sculptures were either lost or disseminated in Venice or elsewhere. The order of the Servants was suppressed by a Napoleonic decree, and all the goods were owned by Napoleon's Kingdom of Italy (on 13 June 1810). On 21 January 1814, the area, including the Basilica, the sacristy, the chapel, and other buildings, was still standing but completely emptied. The church destruction was completed by Nicolò Brazzoduro, Baldassarre Varetton, and Marco Folin, selling the building materials.

Only the Volto Santo chapel was saved because it was used as a stable and warehouse until the end of the nineteenth century [18].

### 1.4. The Main Façade

The main façade is characterized by the typical Venetian combined use of Istrian stone and red Verona marble. The main portal is surrounded by doorposts in Istrian stone and the architrave on which is engraved the inscription commemorating the consecration of 7 November 1491 [18]. The top of the portal is decorated with Proconnesian marble slabs (probably reused from another site due to the imperfect way they have been cut and implemented) and two small *tondos* made of dark red marble (*Cipollino* marble) (Figure 1a). Stones and bricks show exfoliation and material loss (Figure 1b,c), spalling (Figure 1c), and black crusts (Figure 1d), and the presence of moss-grown and secondary plants (Figure 1e).

The Santa Maria dei Servi Church was selected as a case study since it is made of original bricks and shows typical main deterioration effects found in Venice lagoon cultural heritage.

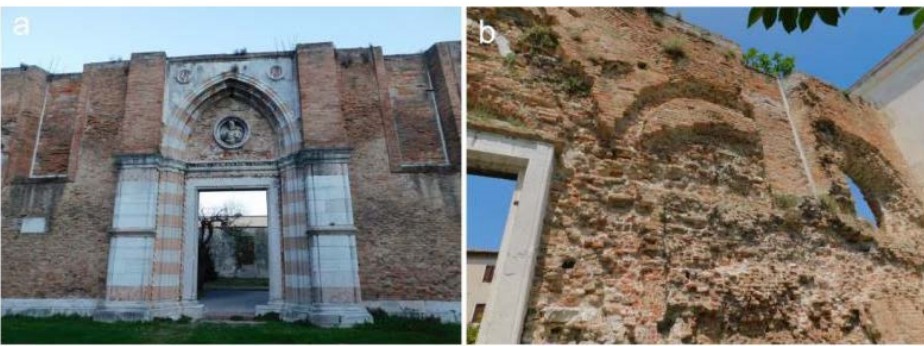

**Figure 1.** *Cont.*

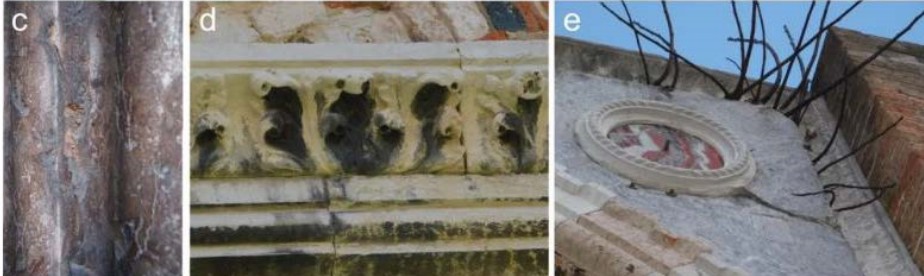

**Figure 1.** Details of the main façade of the ruins of Santa Maria dei Servi Church: (**a**) the external main portal; (**b**) the internal main portal; (**c**) exfoliation of red Verona marble used in the portal; (**d**) detail of ornamental Istrian stone use in the portal with black crusts in sheltered areas; (**e**) detail with Proconnesian marble (grey) and the *tondo* in *Cipollino marble* (red); secondary plants (*ficus carica*) growth is also visible.

## 2. Materials

Twenty-three brick samples (Figure 2) were collected on the main façade of the Basilica of Santa Maria dei Servi. In order to guarantee good representativeness of the materials used and their decay conditions, samples were randomly collected, covering most of the surface of the façade.

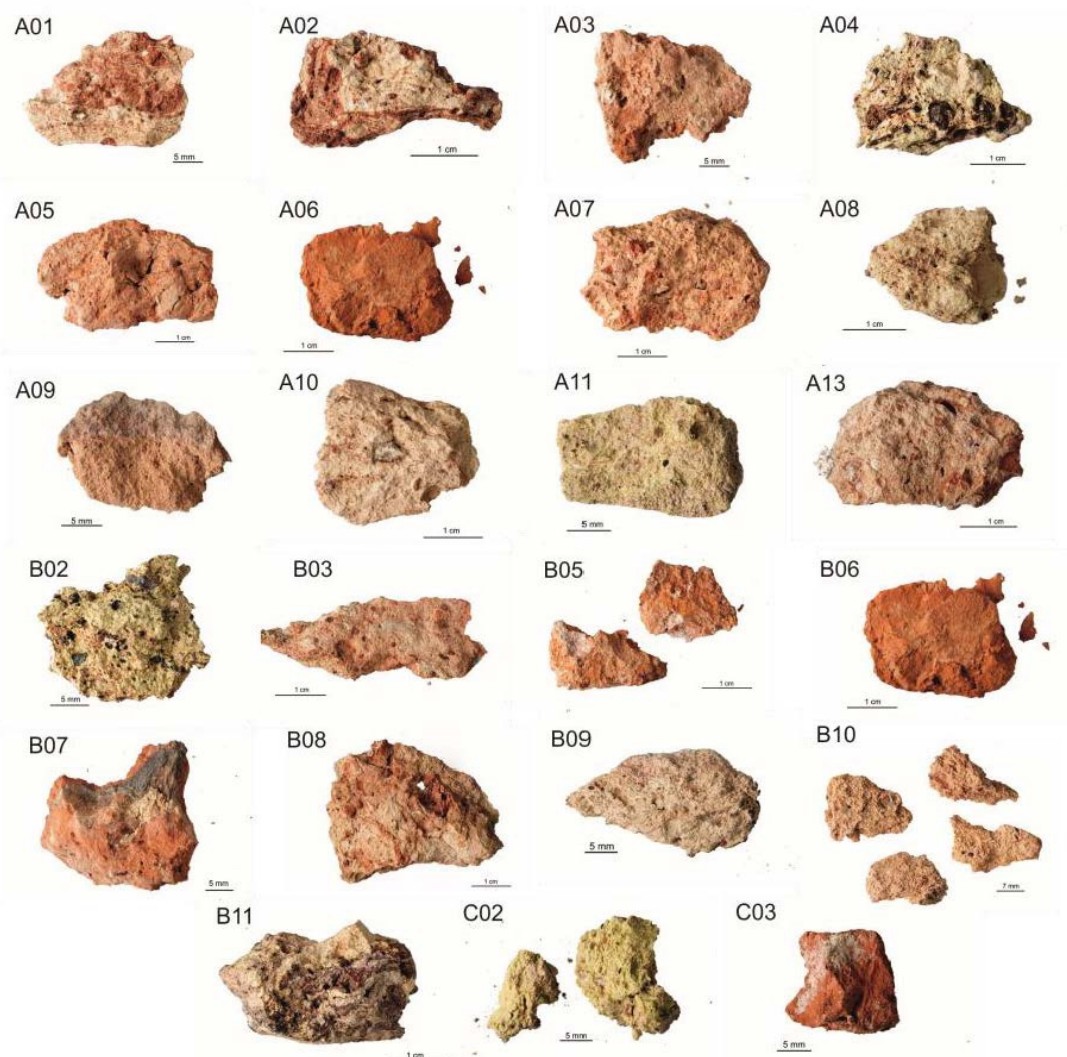

**Figure 2.** Macroscopic photographs of the twenty-three collected samples.

Samples collection was operated in three areas:

(i) side A, the outside masonry on the right of the main portal where eleven samples were collected: A01, A02, A03, A05, A06, A07, A08, A09, A10, A11, A13;

(ii) side B, the outside masonry on the left of the main portal, where nine bricks were collected: B02, B03, B05, B06, B07, B08, B09, B10, B11;

and (iii) side C, the backside of area B, corresponding to the inside of the construction, where two samples (C02 and C03) were selected.

## 3. Methods

The collected samples were analyzed under the mineralogical and textural point of view in the laboratory, using standard methods such as spectrophotometry, X-ray powder diffraction (XRPD), polarized optical microscopy (OM), field emission scanning electron microscopy coupled with an energy dispersive spectrometer (FESEM-EDS) and by hyperspectral analysis (HA) (Table 1). Moreover, the external façade was surveyed by on-site measurements with hyperspectral cameras in order to obtain specific decay maps (biodeterioration and salt presence distribution) of all the studied surfaces from which the samples were collected.

**Table 1.** List of the performed analyses, expected results, and number of the studied brick samples.

|  | Methods | Expected Results | N. Samples |
|---|---|---|---|
| | Spectrophotometry | Aesthetic aspect in dry and wet conditions | 23 |
| | OM | Petrographic study, mineral content and texture/porosity | 23 |
| In lab | FESEM-EDS | Texture, porosity, mineral boundary reactions | 5 |
| | XRD | Mineral composition | 25 |
| | HA | Mineralogy and decay products | 3 |
| On-site | HA | Decay map of the overall main façade | |

The color of dried and wet bricks samples was assayed on a 3 nh spectrophotometer according to the CIELab system, which describes the color as lightness (L*: −100 = black, +100 = white) and adds the two chromatic coordinates a* and b*, which reflect the amount of red-green and yellow-blue colors (a*: −60 = green, +60 = red; b*: −60 = blue, +60 = yellow), respectively. The degree of color difference (ΔE*) was calculated according to the following equation:

$$\sqrt{\left(L_1^* - L_2^*\right)^2 + \left(a_1^* - a_2^*\right)^2 + \left(b_1^* - b_2^*\right)^2}$$

where subscript '1' refers to measurements on dry samples and subscript '2' on wet ones.

Thin sections were examined under a polarized-light optical microscope (Olympus DX-50, equipped with a Nikon D7000 digital microphotography system).

Mineralogy and texture were investigated by a FESEM Tescan Solaris at the department of Geosciences (University of Padua). Microchemical analysis was performed on mineral phases as observed under the FESEM, using an Oxford Instrument Ultim Max 65 Silicon drift detector EDS, standardized with natural silicates and oxides (Amelia plagioclase for Si, Na and Al; diopside for Ca; San Carlos olivine for Mg, orthoclase for K; $BaSO_4$ for sulfur and $Fe_2O_3$, CrO, $MnTiO_2$ and NiO natural oxides), and operating at 15 KeV with a current of 3 nA and a working distance of 5 mm. Backscattered electron (BSE) images were acquired on polished sections to assess the brick texture, working at a lower tension and current (5 KeV, 300 pA and at a working distance of 4 mm) to improve image resolution.

X-ray powder diffraction (XRPD) was performed to identify the mineral phases. Diffraction data were acquired on a PANalytical X'Pert PRO diffractometer, operating in Bragg-Brentano reflection geometry with CuKα radiation, 40 kV of voltage and 40 mA of filament current, equipped with an X'Celerator detector. Qualitative analysis of diffraction data was carried out with X'Pert HighScore Plus® software (PANalytical) and the PDF-2

database. The semi-quantitative concentration of minerals was estimated by the mass fractions of the accepted phases using the RIR (Reference Intensity Ratio) values from the database used to perform the analysis.

Hyperspectral analyses were carried out both in the laboratory (on sampled bricks) and on the field (on-site). Laboratory analyses were useful in creating a database of the spectral signature of the original un-weathered materials. In addition, hyperspectral data were also acquired on experimental alteration samples (tests) created in the laboratory (e.g., for salt crystallization or exposure to atmospheric conditions) in order to have data on the possible effects of specific conditions on the different materials and to identify the diagnostic absorption bands.

Laboratory analyses were done with the Headwall Photonics Nano-Hyperspec and Micro-Hyperspec cameras. These hyperspectral cameras work in the VNIR range (Visible and Near InfraRed, 400–1000 nm) and in the SWIR range (Short Wave InfraRed, 900–2500 nm), respectively. The Nano-Hyperspec camera has a total of 270 spectral bands and 640 pixels per slit, and the spatial resolution used in our laboratory setup was around 0.15 mm. The Micro-Hyperspec camera has a total of 166 spectral bands and 384 pixels per slit, and the spatial resolution used in the laboratory was around 0.35 mm.

The on-site survey was done with the Nano-Hyperspec camera using a tripod at a distance of 5 m, giving a resolution of around 3 mm with the VNIR camera and around 5 mm with the SWIR camera. To detect the biodeterioration, we mapped the 700 nm band distinctive of chlorophyll using the equation $1 − (b1/((b2 + b3)/2))$, where b1 is the chlorophyll band center (678 nm) and b2 and b3 the shoulders of the band (653 nm and 706 nm, respectively). Regarding the sulfates, it was not possible to map their distribution on the building using the same method since their main distinctive features in the SWIR range were obliterated by the atmospheric absorption. For this reason, we used two spectral indexes, which ignored those obliterated spectral ranges. In particular, we used the equations from Shuai et al. (2022) [21]. The two spectra indexes are: $b4 \times b5/b6 \times b7$ and $b4 \times b5/b7 \times b7$, where b4 is the 1660 nm band, b5 the 2210 nm band, b6 the 2260 nm band and b7 the 2340 nm band [21].

## 4. Results and Discussion

### 4.1. Color Measurements

According to colorimetric results in the Lab space (Table 2), bricks (in dry condition) can be defined into three main chromatic groups (Figure 3): yellow-colored bricks (group1_yellow), pink-colored bricks (group2_pink), and red-colored-bricks (group3_red). The most abundant category of bricks was the pink-colored (with twelve samples), followed by the red- (seven samples) and yellow-colored ones (four samples).

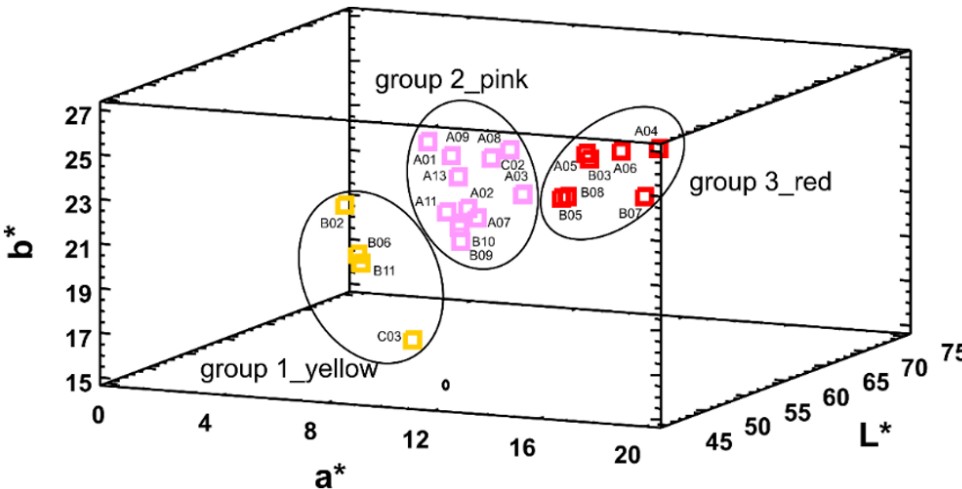

**Figure 3.** Colorimetry: dried bricks in Lab space. L*, a*, b* = color coordinates.

**Table 2.** Color coordinates (L*, a*, b*) of bricks samples measured in dry and wet conditions, and color difference (ΔE) derived by dry and wet samples. Classification in the three groups according to Figure 3: group1_yellow; group2_pink; group3_red.

| Sample | DRY | | | WET | | | ΔE | Group |
|--------|------|-------|-------|------|-------|-------|-------|-------------|
| | L* | a* | b* | L* | a* | b* | | |
| A01 | 61.40 | 11.64 | 23.40 | 51.80 | 12.38 | 24.98 | 9.76 | group2_pink |
| A02 | 62.28 | 7.89 | 20.95 | 49.24 | 12.50 | 25.51 | 14.56 | group2_pink |
| A03 | 56.59 | 11.70 | 22.65 | 39.96 | 19.82 | 24.49 | 18.60 | group2_pink |
| A04 | 52.44 | 18.04 | 25.84 | 44.89 | 17.88 | 26.17 | 7.56 | group3_red |
| A05 | 59.00 | 13.35 | 24.33 | 57.69 | 11.48 | 26.97 | 3.50 | group3_red |
| A06 | 67.07 | 8.28 | 23.24 | 49.32 | 10.60 | 23.65 | 17.91 | group3_red |
| A07 | 60.95 | 8.64 | 20.76 | 55.78 | 12.51 | 27.10 | 9.05 | group2_pink |
| A08 | 66.25 | 8.33 | 23.07 | 58.62 | 7.81 | 27.33 | 8.76 | group2_pink |
| A09 | 66.33 | 5.21 | 23.16 | 49.97 | 13.95 | 24.94 | 18.63 | group2_pink |
| A10 | 70.42 | 5.24 | 18.96 | 60.72 | 11.15 | 28.20 | 14.64 | group2_pink |
| A11 | 67.25 | 9.63 | 19.89 | 56.60 | 6.20 | 27.59 | 13.59 | group2_pink |
| A13 | 45.23 | 12.60 | 25.05 | 51.41 | 15.97 | 26.80 | 7.26 | group2_pink |
| B02 | 59.67 | 4.06 | 21.10 | 51.04 | 14.49 | 35.47 | 19.74 | group1_yellow |
| B03 | 62.12 | 12.56 | 23.58 | 49.81 | 16.13 | 26.36 | 13.12 | group3_red |
| B05 | 58.68 | 12.54 | 22.27 | 43.47 | 17.34 | 24.66 | 16.12 | group3_red |
| B06 | 64.49 | 3.08 | 18.15 | 56.05 | 3.83 | 21.98 | 9.30 | group1_yellow |
| B07 | 50.20 | 18.17 | 23.96 | 36.81 | 23.45 | 24.78 | 14.42 | group3_red |
| B08 | 58.99 | 12.66 | 22.31 | 45.15 | 17.68 | 25.27 | 15.03 | group3_red |
| B09 | 64.79 | 6.89 | 19.06 | 55.42 | 10.88 | 24.31 | 11.46 | group2_pink |
| B10 | 60.38 | 8.17 | 20.34 | 51.95 | 10.26 | 23.29 | 9.17 | group2_pink |
| B11 | 57.65 | 5.13 | 19.13 | 40.69 | 9.37 | 18.46 | 17.49 | group1_yellow |
| C02 | 63.74 | 2.36 | 27.99 | 55.08 | 3.03 | 33.42 | 10.25 | group2_pink |
| C03 | 61.13 | 8.16 | 15.05 | 41.31 | 16.04 | 20.55 | 22.03 | group1_yellow |

Color changes in dry and wet samples were measured in order to verify esthetical qualities in different microclimate conditions (presence of rain and humidity). The colorimetry results obtained on the dry and wet samples showed slight differences. All samples resulted in being sensitive to color changes after wetting: in general, a* and b* (yellow-red and blue-green components, respectively) increased, and L* (lightness) decreased (Table 2).

In general, all samples seemed to be sensitive to color changes, as the value ΔE showed (in most of the samples ΔE is >10). ΔE had a maximum difference of 22.03 (sample C03) and a minimum of 3.50 (sample A05). In general, wet samples increased in b* and a* parameters and decreased in lightness (L*), with few exceptions like brick A12 and A13. All the color changes were over the limit of perceptibility, defined as a $\Delta E \geq 3$ in the CIELAB color space [22], and therefore are visible to the naked eye [23].

### 4.2. Texture and Mineralogy

#### 4.2.1. Petrographic Analysis

Under a textural viewpoint, bricks displayed similar features, with a very low optically active groundmass, with color ranging between red or red-brown and yellow-light to brown (Figure 4a–c). Most samples (such as A10, A11, A13, B02, B06, B10, and C02) had a compact matrix, optically inactive, showing highly dense gelling portions (Figure 4c,d). Most samples had planar voids and vughs (Figure 4e), but the channel and vesicular pores were also present where clay pellets occurred (Figure 4f). Most inclusions (~30%) observed were grains of quartz and subordinately feldspar. Clay pellets were rather abundant (20%). Residual carbonates (calcite and/or dolomite) (Figure 4g) suggested that temperatures did not exceed 800 °C (such as in sample B05), a temperature over which they are completely decomposed [3,24,25]. The inactive optical groundmass in other samples (such as B02, Figure 4d) suggested the reaching of high temperatures (>1000 °C) during firing since phyllosilicates and carbonates in the matrix (fine-grained) lost all their interference colors

when they broke down. However, carbonate relics testified to the occurrence of coarse calcite grains in the clay paste, which did not completely decompose during the firing process due to kinetic reasons (too rapid heating rate during firing and/or short soaking times). Moreover, secondary calcite could be derived from the carbonation of lime "lumps" formed during the firing of clays containing carbonate grains (calcareous clays) [26]. Post-depositional calcite was easily recognizable since it tended to partially fill pores (Figure 4h), even if gypsum crystals were also detected as weathering product filling pores and/or at the surface of the sample (where it was directly in contact with the lagoon environment) (Figure 4i).

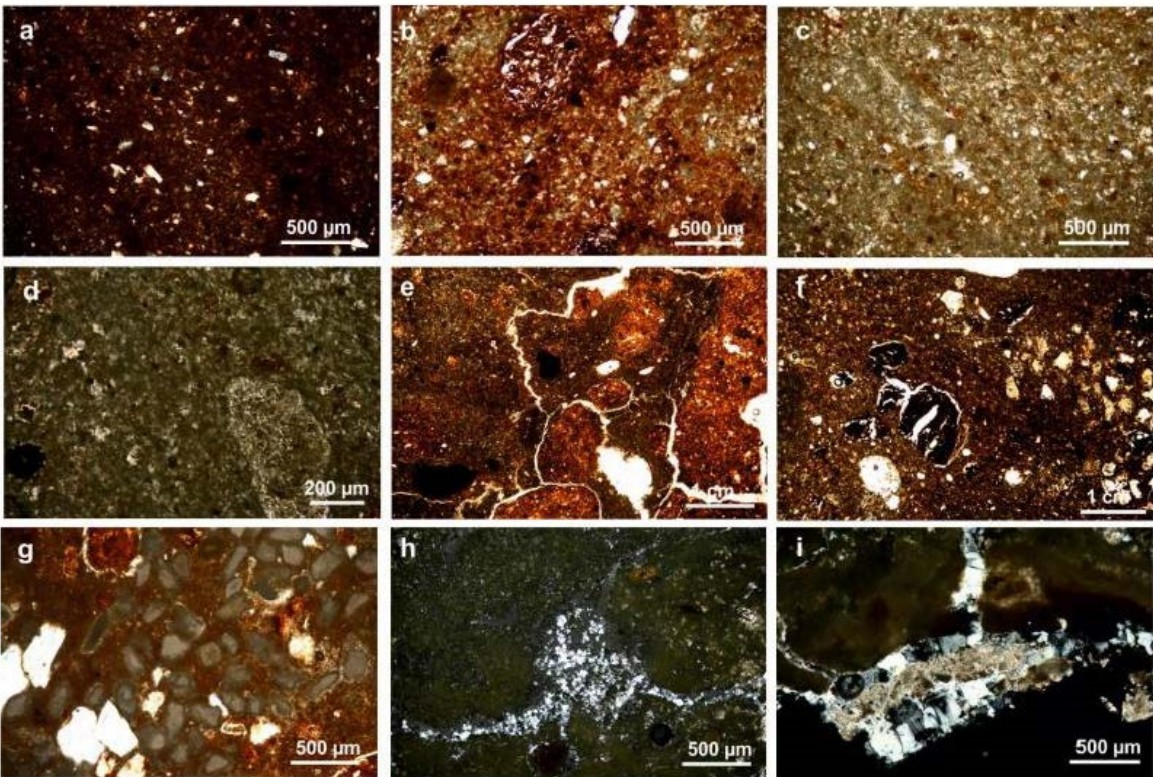

**Figure 4.** Photomicrographs in plane-polarized light (PPL) and crossed-polarized light (XPL) of bricks collected from Santa Maria dei Servi. (**a**) sample B07 (PPL): red-brown groundmass; (**b**) sample B08 (PPL): pale red groundmass; (**c**) sample B11 (PPL): yellow groundmass; (**d**) sample B02 (PPL): slightly active groundmass; (**e**) sample B05 (PPL): planar voids vesicles; (**f**) sample B08 (PPL): planar voids and vesicles; (**g**) sample B05 (PPL: grains of carbonate relicts; (**h**) sample C02 (XPL): secondary calcite filling pores); (**i**) sample C02 (XPL): gypsum crystals in pores and at surface materials.

4.2.2. Micro-Textural and Micro-Chemical Analysis

Further micro-textural observations were carried out under FESEM on a set of six samples (A04, A08, B02, B06, B10, and C03). BSE images confirmed what was drawn by petrographic characterization. Samples that displayed a high sintered matrix (samples A04, A08, B02, B06, B10) were characterized by a wide pore texture, with a prevalence of micro- and meso-rounded pores (vesicles) (Figure 5a,b) and pervasive phenomena of bridging interconnection (Figure 5c) in particular between residual grains of quartz and feldspar. Groundmass was highly dense (sintered) and reacted (Figure 5d) and raw minerals showed different states of decomposition according to the dimension of the grains, often with reaction rims where new silicates were formed [3,24,27,28] (Figure 5d). Carbonate relicts were also present (Figure 5a), although the high degree of textural evolution which suggested high temperature was reached during firing. Prismatic secondary minerals were also dispersed in the sintered matrix, mainly composed of Mg-rich silicates (diopside).

Meanwhile, Mg- Ti- Fe- rich pyroxenes (augite) were observed by the higher atomic density (chemical analysis was performed by EDS) formed [29] on reaction rims of feldspars (Figure 5e,f) or titanite minerals (Figure 5g). The presence of Ti-minerals suggested the use of lagoon sediments [30], mainly characterized by quartz, K-feldspar, and plagioclase, and heavy elements such as titanium. Iron-bearing phases also occurred in residual illite, where grains partially underwent melting and presented typical "bubble-like" porosity. However, illite still showed dehydroxylated basal sheets, such as in sample C03 (Figure 5h). This sample was characterized by the lowest textural evolution, fewer new silicate phases, a low interconnection between minerals, and jagged pores (Figure 5i) [10].

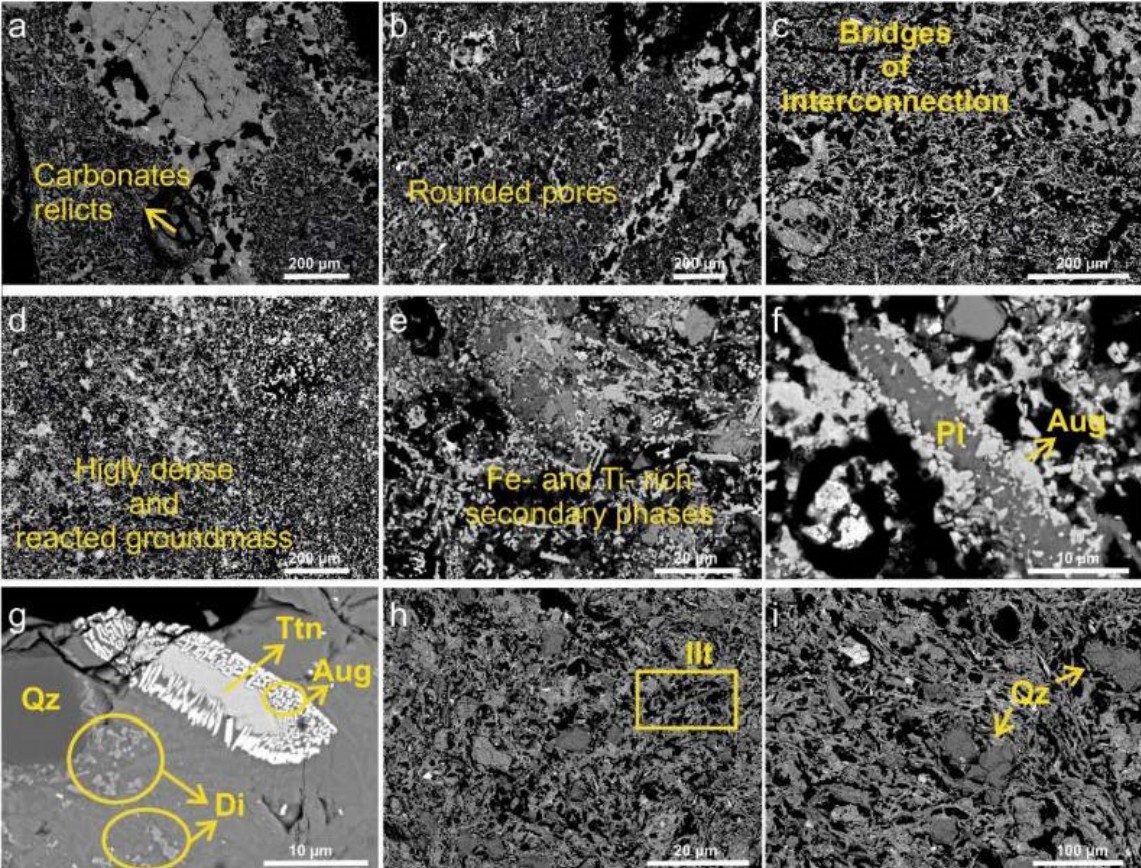

**Figure 5.** FESEM-BSE images of micro-texture of the bricks and minerals detected in the groundmass: (**a**) sample A08, secondary porosity (in black) and relict of carbonates; (**b**) sample A04, micro-texture characterized by rounded large pores (vesicles) and silicates crystalized during firing (high temperature silicates); (**c**) sample B02, bridges of interconnection between reacted grain minerals; (**d**) sample B06, high reacted groundmass and diffuse presence of newly formed minerals; (**e**) sample B06, new minerals formed during the firing process, mainly Fe- and Ti- rich silicates; (**f**) sample B06, augite (Aug) along feldspar (Pl, plagioclase) rims; (**g**) sample A08, new silicates, titanite (Ttn), augite (Aug) and diopside (Di); (**h**) sample C03, illite (Ilt) relicts in a poorly sintered groundmass; (**i**) sample C03, unreacted quartz (Qz) grains embedded into a poorly sintered groundmass and low micro-textural evolution.

### 4.2.3. Mineralogy and Decay Products

The XRPD analysis showed a relative abundance of primary minerals (i.e., illite, calcite, and dolomite) and those formed during firing (i.e., diopside, gehlenite), suggesting a certain homogeneity in raw materials used and a no-standardization in the firing process (in particular concerning the firing temperature reached). The main mineral phases found can be divided into: (i) minerals occurred in the raw materials that did not completely react

during firing, such as quartz (Qz), K-feldspar (Kfs), plagioclase (Pl), calcite (Cal), dolomite (Dol), illite (Ilt); (ii) new minerals formed during firing, such as diopside (Di), gehlenite (Gh), augite (Aug), K-feldspar (sanidine), and hematite (Hem); and (iii) decay products formed after the brick was settled in the building, such as gypsum (Gyp), aluminite (Alu), mirabilite (Mir), and analcime (Anl) (see Table 3).

**Table 3.** Mineralogical assemblages in historical bricks defined by XRPD analysis and semi-quantitative analysis by RIR. Mineral abbreviations after Whitney and Evans (2010) [31]: Qz = quartz; Kfs = K-felspar; Pl = plagioclase; Cal = calcite; Dol = dolomite; Ilt = illite; Di = diopside; Gh = gehlenite; Aug = augite; Fo = forsterite; Gyp = gypsum; Alu = aluminite; Anl = analcime; Mir = mirabilite; Hal = halite; Hem = hematite. Am = amorphous. Amorphous phase estimated abundance (based on the diffraction pattern): xx = abundant; x = medium; + = scarce.

| | Qz | K-fs | Pl | Cal | Dol | Ilt | Di | Geh | Aug | Fo | Gyp | Alu | Anl | Mir | Hal | Hem | Am |
|---|---|---|---|---|---|---|---|---|---|---|---|---|---|---|---|---|---|
| A01 | 45 | 12 | 23 | | 3 | 1 | 9 | | | | 4 | | | | | 3 | |
| A02 | 23 | 16 | 33 | | | | 23 | 2 | | | | | | | | 3 | |
| A03 | 22 | 33 | 22 | 5 | 4 | | 12 | | | | | | | | | 2 | |
| A04 | 13 | | 25 | 5 | | | 19 | | 22 | | 2 | | 10 | 1 | 1 | 2 | xx |
| A05 | 35 | 27 | | 5 | 6 | 8 | | | | | 7 | 9 | | | | 3 | x |
| A06 | 27 | 16 | 17 | 10 | | 12 | 9 | | | | 3 | 3 | | | 1 | 2 | |
| A07 | 6 | 21 | 33 | | | | 15 | 3 | 13 | | 4 | | 3 | | 1 | 1 | |
| A08 | 9 | 12 | 13 | 6 | | | 28 | | 15 | | 3 | | 11 | | 1 | 2 | xx |
| A09 | 28 | 16 | 28 | | | | 19 | | | 6 | | | | | | 3 | |
| A10 | 7 | 11 | 24 | 4 | | | 28 | | 17 | | 2 | | 6 | | | 1 | |
| A11 | 5 | | 22 | | | | 29 | | 28 | | 4 | | 11 | | | 1 | xx |
| A13 | 25 | 18 | 25 | | 4 | | 13 | | | | 14 | | | | | 1 | |
| B02 | 3 | 8 | 33 | | | | 16 | | 21 | | 6 | | 11 | | | 2 | xx |
| B03 | 27 | 16 | 33 | | | | 11 | | 12 | | | | | | | 1 | + |
| B05 | 31 | 19 | 12 | 11 | 12 | | | | | | 12 | | 2 | | | 1 | |
| B06 | 12 | 11 | 18 | | | | 19 | | 16 | | 6 | | 15 | | | 3 | xx |
| B07 | 42 | 15 | 12 | 8 | 6 | 8 | | | | | 4 | | | 1 | 1 | 3 | |
| B08 | 25 | 22 | 23 | | | | 15 | | 13 | | | | | | | 2 | |
| B09 | 28 | 17 | 24 | | | | 12 | | 6 | | 8 | | 3 | | | 2 | x |
| B10 | 21 | | 22 | | | | 15 | | 15 | 7 | 14 | | 3 | 1 | 1 | 1 | x |
| B11 | 19 | 15 | 18 | | | | 11 | | 11 | | 22 | | 3 | | | 1 | |
| C02 | 11 | | 25 | | | | 25 | 4 | 12 | | 21 | | | | | 2 | xx |
| C03 | 27 | 19 | 16 | 8 | 1 | | 13 | | | | 9 | 2 | | 1 | 2 | 2 | |

The predominant mineral phase was quartz; even in a few exceptions, it was still present in low/medium quantities (A04, A08, B02, C02). Quartz was followed by feldspars (plagioclases and K-feldspars) which were also abundantly detected in almost all the samples (Table 3). Samples had the characteristic peaks of calcite, and some of them also had dolomite and illite (A01, A05, A06, B07, and C03). The presence of these phases indicated that the firing temperature did not exceed 750 °C (when dolomite is present) or 800–850 °C (when only calcite is occurring) [25,27]. It is important to consider that calcite starts decomposing at around 700 °C, but it can be detected up to 800/900 °C [32,33] since decarbonation temperature also depends on the grain size as well as firing condition (oxidizing or reducing). In addition, dolomite was detected in a fair number of samples, sometimes associated with calcite (e.g., A03, B07, C03). Dolomite starts to decompose at 700 °C, according to a two-step process as described below [32]:

$$CaMg(CO_3)_2 \Leftrightarrow CaCO_3 + MgO + CO_2$$

$$CaCO_3 \Leftrightarrow CaO + CO_2$$

This different behavior during firing could explain why we found only calcite instead of both of them and why diopside was in general very abundant in many samples (A04, A08, A10, A11, B02, C02), in particular where carbonates were totally absent (B02

or C02). The calcite detected may also largely correspond to secondary calcite, a result of the re-carbonation of unreacted CaO (such as already observed by FESEM). For this reason, illite can also be used to constrain the firing temperature to below 900–950 °C [3]. K-feldspar also progressively disappeared as temperature increased, or it could be transformed into high-temperature polymorphs (sanidine) or reacted to form new minerals (e.g., anorthite) [32]. Other Ca-, Mg-, and Ti-silicate phases were also detected, indicating that the firing process reached high temperatures. Diopside frequently occurred as a result of the reaction between dolomite and quartz and started to form at about 850 °C [32]. Nevertheless, the presence of diopside and carbonates in the same diffractograms (e.g., A13, C03) indicated temperatures reached between 800 and 900 °C, meaning that an incomplete decarbonation and a formation of new minerals process might still have places [3,25,32,34]. Augite was also attested in a few bricks, as a result of reactions in raw materials containing abundant iron or titanium, according to the FESEM analysis. Forsterite was only noted in some bricks (samples A09 and B10). The coexistence of carbonates, illite, and new silicate phases suggested the absence of good standardization in the firing process. This supported the theory that in the 14th century there was an initial phase of local production of bricks using local clay of the lagoon and without standardization in terms of the quality of the final products [14].

Salts and sulfates (halite, mirabilite, and gypsum) were often found by diffraction patterns. These results are consistent with previous literature data, which indicated gypsum, halite, and mirabilite as the main weathering products due to the salt decay process that affects materials used in Venice buildings and the lagoon environment [16,17]. Their presence is related to the phenomena of the migration of saline solutions from the ground and of the sea spray. Gypsum is a decay product of the reaction between today's urban atmosphere and building materials (both bricks and mortars), and it is especially ascribed to the different reactivity of atmospheric sulfur dioxide [27,35]. Water migration also causes a general dissolution of the salt inside the wall, and when evaporation takes place, their crystallization occurs on the surface [2]. Analcime (probably ascribed as Mg- and Ca-zeolites) was also detected in many samples, in particular in those presenting high amorphous phases (Table 3). It agrees with the weathering process of a highly sintered matrix, which forms zeolites as secondary products [26,28]. Although halite was detected in a few samples (Table 3), it is the main cause of aloclastism in brick masonry in the lagoon environment. For this purpose, a sample of salt efflorescence was collected (by Santa Fosca Church in Cannaregio, nearby Santa Maria dei Servi Church) and analyzed, confirming the presence of halite (see Supplementary Materials, Figure S2). This is in accordance with the climate of Venice, where the dissolution and crystallization of the salts is one of the most common weathering phenomena.

Hierarchical cluster analysis carried out on diffraction patterns [36] allowed the statistical grouping of samples according to the peak position and intensity. Thus, bricks were grouped into five different clusters (Figure 6a), emphasizing differences between samples in terms of their firing temperatures as well as their weathering products. Cluster 3 and Cluster 5 grouped bricks fired at the lowest temperature, probably below 750 °C, marked in particular by the presence of illite and dolomite. They had the most representative brick of the set, samples A05 and B07, respectively (see diffractograms in Figure 6b), and they were distinguished by the peak (~7.79 Å, d-spacing) of Al sulfate–hydroxide minerals aluminite, $(Al_2SO_4(OH)_4 \cdot 7H_2O)$, weathering product attributed to the natural reaction of acidic sulfate with clay material [37]. Clusters 1 and 2 grouped bricks fired at a high temperature (above 950 °C). These are bricks that displayed a high density of matrix in optical microscopy and high micro-textural and mineralogical evolution by BSE analysis. Diffractograms of samples B06 and B09 (representative samples of Clusters 1 and 2, respectively) showed indeed the absence of carbonates, abundant new silicates (in particular from the pyroxene group, calcium-, magnesium-, and iron/titanium- rich varieties), and the absence of amorphous phases. These two groups, similar in terms of firing secondary phases, were distinguished by the presence of zeolite (analcime) as the weathering product,

whose peaks are predominant in Cluster 1. Cluster 4 is intermediate. It identified bricks probably fired between 800 and 950 °C since when illite disappeared, new mineral phases were formed, and diffractograms did not show amorphous bands. Moreover, this set of bricks also did not show weathering products.

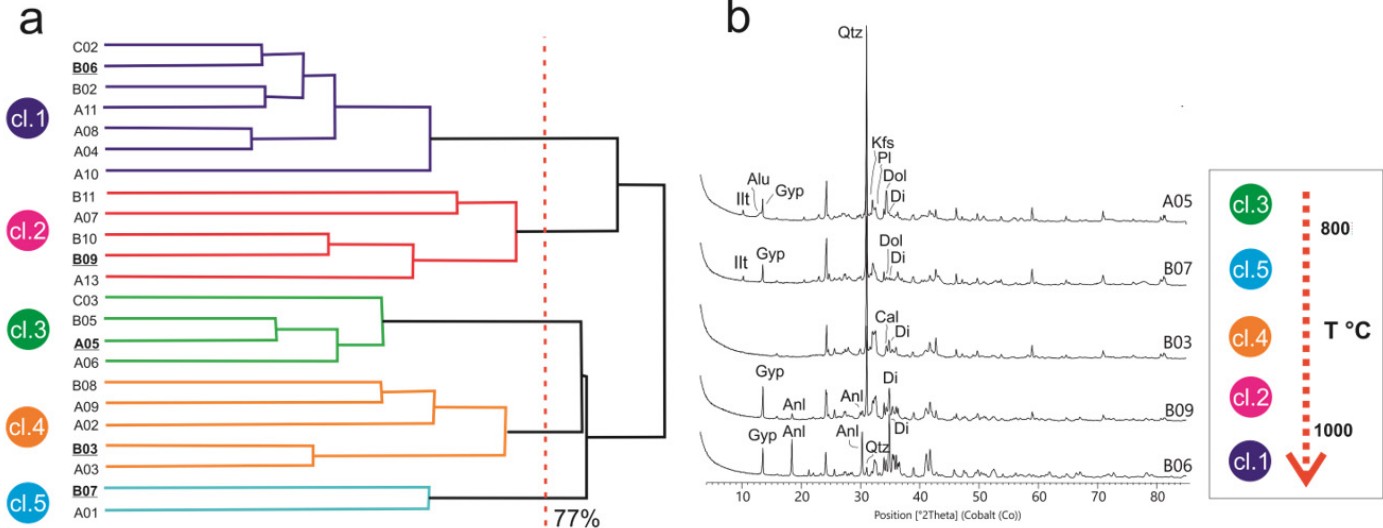

**Figure 6.** Hierarchical cluster analysis of XRPD data: five clusters grouped all samples according to peaks position and peak intensity: (**a**) dendrograms with clusters in different colors and the most representative samples were bolded. The red dotted line corresponds to the cut-off, defining the clusters; (**b**) comparison between diffractograms of the most representative samples of each cluster according to the firing temperatures defined on the basis of mineralogical associations detected by XRPD analysis.

*4.3. Mapping of Deterioration Products*

Laboratory tests were carried out on three samples (C03, B08, and B05) (Figure 7) in the laboratory by HA in order to define the dataset for secondary phases and biodeterioration identification to be then applied for the on-site (for all the monument surfaces here studied) mapping.

In agreement with XRPD, SWIR spectral maps highlighted areas rich in hydrated sulfates (e.g., gypsum or mirabilite), with the typical triplet absorption bands at ~1440 nm, ~1500 nm, and ~1540 nm and the band at 1750 nm (Figure 7). In the VNIR range, the most indicative band was the one related to chlorophyll, a tracer of biodeterioration, whose band center is at around 680 nm.

In the on-site hyperspectral acquisitions of the façade of Santa Maria dei Servi Church, SWIR ranges unfortunately were not observable due to the atmosphere absorption between 1334–1439 nm and between 1794–1938 nm. The 1750 nm band was completely obliterated, while the triplet between 1440–1540 nm showed some artifacts due to the close relationship with the 1334–1439 nm atmosphere absorption. For this reason, we used the spectral indexes proposed by Shuai et al. [21]. The two spectral indexes gave identical results, showing enrichment in sulfates in the lower part of the wall likely due to the capillarity rise of water (Figure 8a,b). A slightly higher concentration of sulfates was also seen on the top of the wall, especially when compared to the central part (Figure 8b). In this case, the enrichment is probably related to meteoric water and sea spray present in the lagoon environment. The on-site chlorophyll spectral maps (Figure 8c,d) showed a quite random distribution of biodeterioration. The chlorophyll band was a bit deeper in the lower part (closer to the ground) and the brick cracks, indicating a little enrichment in these areas.

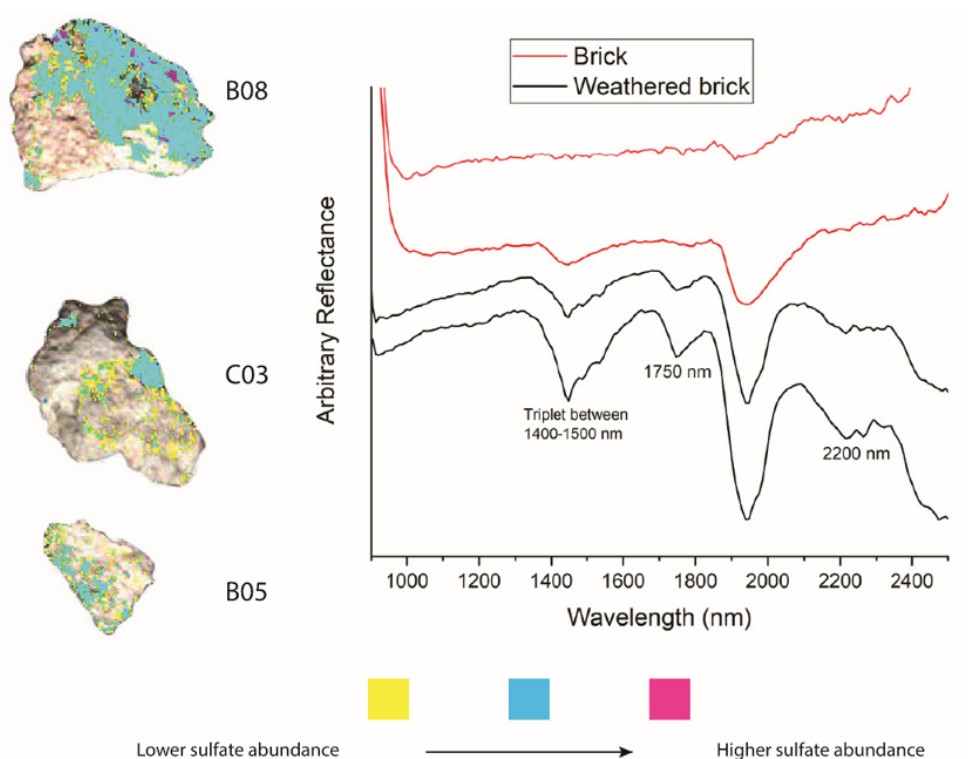

**Figure 7.** Spectral maps of the three fragments (Figure 2) of the Santa Maria dei Servi bricks show the sulfate abundance. The spectra are relatively featureless for un-weathered bricks (with only the 1900 nm band indicative of hydration). On the contrary, the weathered portions show a spectrum with distinctive hydrated sulfate bands (triplet between 1400–1500 nm, 1750 nm, and 2200 nm).

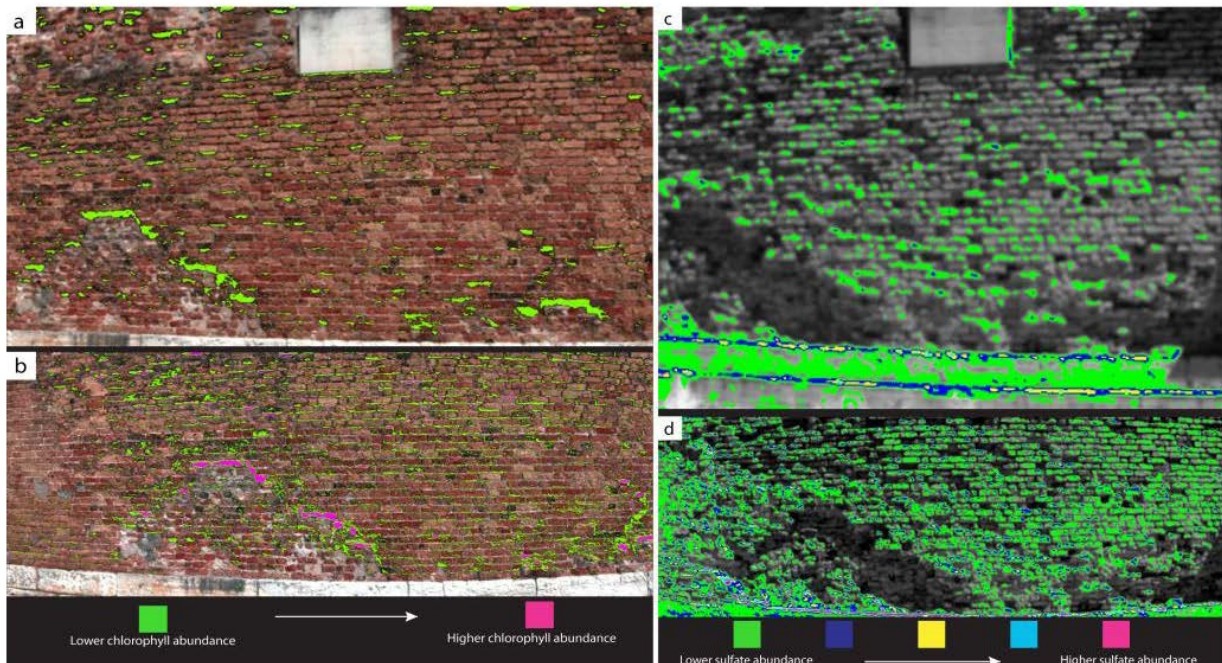

**Figure 8.** The on-site hyperspectral acquisition of the façade of Santa Maria dei Servi Church: (**a**) low-resolution and (**b**) high-resolution sulfate spectral maps based on the Shuai et al. [21] spectral index; (**c**) low-resolution spectral map showing the general trend of biodeterioration; (**d**) high-resolution map of the chlorophyll absorption band. Deeper chlorophyll band depth reflects a higher concentration of biodeterioration, highlighted in pink.

## 5. Conclusions

The methodological approach adopted on a set of twenty-three brick samples collected on the main façade of Santa Maria dei Servi Church allowed to identify the raw materials used, constrain the firing process adopted, in particular concerning the maximum temperature reached, and recognize the main weathering products caused by the Venetian lagoon environment.

Results showed that:

1. Petrographic and microstructural analysis by FESEM-EDS analysis was extremely useful in studying the microtextural and the mineralogical evolution of the collected bricks. Pore shape and dimensions, texture, and mineralogy detected suggested that temperatures reached during the firing process ranged between 700–1000 °C. The same samples displayed incomplete mineral reactions, while others had a high density and a wide presence of new silicates formed during firing. The presence of Ti-phases supported the theory that in the 14th century there was an initial phase of local bricks production using the local clay of the lagoon.

2. XRPD analysis confirmed the presence of carbonates, illite, and new Ca-, Mg-, and Ti-silicate phases. The presence of these mineral phases and their abundances were important indicators not only for knowing the mineralogical composition of the raw materials used but also for understanding the temperature reached during the firing process. The raw materials used contained quartz, K-feldspar, plagioclase, as well as carbonates (calcite and dolomite). Gehlenite, diopside, and augite were secondary products of the firing reactions at T > 800 °C. The amorphous phase was present in a large number of samples (Clusters 1 and 2 in Figure 6). Gypsum was often found as a decay product. Meanwhile, zeolite (analcime) was found as a weathering product of samples that displayed more amorphous in XRPD patterns.

3. The poor homogeneity in the mineralogical evolution among the collected bricks suggested the absence of good standardization in the firing process, probably in relation to a large number of materials to be fired in the kilns and the piling density. This result is perfectly in line with the technological possibility of that time, considering that the first public furnaces are dated back to 1327 [14], just three years before the beginning of the construction of the Santa Maria dei Servi Church.

4. Regarding the weathering products, gypsum was found in many bricks' diffraction patterns as well as in the hyperspectral SWIR maps in which sulfates resulted in covering most of the surface of the façade. Thus, the deterioration effects were not only caused by the capillary rise by also the sea spray, which is dominant on the external walls of the historical heritage of Venice.

5. Biodeterioration was also mapped using hyperspectral VNIR ranges. Chlorophyll spectral bands were mapped in all the façade, in particular along mortar joints and exposed brick surfaces.

This work provided meaningful information on brick production in the 14th century, a period in which are attested to the first local furnaces in Venice. Indeed, for the history of the Church, it still has the original bricks, a rare case in Venice where bricks were usually replaced during times since their high deterioration in the salt environment. It also aimed to give back the current situation of the brick masonry of the Santa Maria dei Servi Church, in particular of the main façade, in order to sustain the renewed interest that the academic community has recently shown toward the Servi's complex. Thus, results aimed to support appropriate knowledge of the historical brick manufacturing in Venice and, therefore, to address appropriate interventions in case of restoration or substitution of bricks, or where possible, identify the best solution to prevent the decay.

Results can be applied in all the lagoon environments in Venice, but also the minor islands, such as Torcello, Burano, Murano, and the dry lands in Chioggia e in the historical center of Mestre.

**Supplementary Materials:** The following supporting information can be downloaded at: https://www.mdpi.com/article/10.3390/heritage6020070/s1, Figure S1: High tides in Venice; Figure S2: Salt efflorescence in Venice.

**Author Contributions:** Conceptualization, C.C., L.P.C. and C.M.; methodology, C.C., J.N. and C.M.; investigation, C.C., J.N. and L.P.C.; data curation, C.C., L.P.C. and J.N.; writing—original draft preparation, C.C., L.P.C. and J.N.; writing—review and editing, C.C., L.G., L.M., M.M. and C.M.; project administration, C.C. and C.M.; funding acquisition, C.M. All authors have read and agreed to the published version of the manuscript.

**Funding:** This research was funded by Hyperion project, European Union's Horizon 2020 research and innovation program, grant number 821054.

**Data Availability Statement:** Not applicable.

**Acknowledgments:** The authors thank the Superindence of Venice, Arch. Marco Zordan, Arch. Gianmario Guidarelli (Department of Civil, Environmental and Architectural Engineering—ICEA, University of Padova) and the Santa Fosca Institute for kindly allowing the samples collection and supporting the on-field research activity.

**Conflicts of Interest:** The authors declare no conflict of interest.

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
