# Peer review of "Deterioration Effects on Bricks Masonry in the Venice Lagoon Cultural Heritage: Study of the Main Façade of the Santa Maria dei Servi Church (14th Century)"

_heritage, doi:10.3390/heritage6020070_

Round 1

Reviewer 1 Report

This study presented all the relevant aspects of brick deterioration in an old church in Venice lagoon. Some comments to improve the study follow.

-        Please define all the abbreviations at first mention (SWIR, VNIR, BSE…).

-        Please, try to improve the quality of the figures since they are somewhat blurry. The text in Figure 5 should be clearer.

-        Minerals are to be written with a small capital letter, please change Augite to augite.

-        It would be useful to calculate the semi-quantitative concentration of the detected minerals.

-        It would be beneficial to determine the chemical composition of the samples or some from similar groups.

Author Response

#reviewer1

This study presented all the relevant aspects of brick deterioration in an old church in Venice lagoon. Some comments to improve the study follow.

-        Please define all the abbreviations at first mention (SWIR, VNIR, BSE…).      

I deleted the terms SWIR and VNIR in the abstract, reporting a general Hyperspectral Analysis and HA. Moreover, I checked all abbreviations in the text, and now are defined at the first mention.

-        Please, try to improve the quality of the figures since they are somewhat blurry. The text in Figure 5 should be clearer.

The figures included the text are in high resolution, anyway I agree in the pdf creation they lost their quality. I will check with the editor the possibility to add figures in another way (e.g. sending the original images) in order to maintain the resolution.  

I improved the caption of Fig. 5, that now is: “Figure 5: FESEM-BSE images of micro-texture of the bricks and minerals detected in the ground-mass: a) sample A08, secondary porosity (in black) and relict of carbonates; b) sample A04, mi-cro-texture characterized by rounded large pores (vesicles) and silicates crystalized during firing (high temperature silicates); c) sample B02, bridges of interconnection between reacted grain min-erals; d) sample B06, high reacted groundmass and diffuse presence of newly formed minerals; e) sample B06, new minerals formed during the firing process, mainly Fe- and Ti- rich silicates; f) sample B06, augite (Aug) along feldspar (Pl, plagioclase) rims; g) sample A08, new silicates, titanite (Ttn), augite (Aug) and diopside (Di) ; h) sample C03, illite (Ilt) relicts in a poorly sintered groundmass; i) sample C03, unreacted quartz (Qz) grains embedded into a poorly sintered groundmass and low micro-textural evolution.”

-        Minerals are to be written with a small capital letter, please change Augite to augite.

Thank you. I changed Augite to augite in the caption of Table 3 and I also checked all the manuscript.

-        It would be useful to calculate the semi-quantitative concentration of the detected minerals.     

We agreed with the reviewer but being the collected sample are small due to sampling limitation required by the Venetian superintendence. Such sizes did not allow to perform XRPD with an internal standard for quantitative analysis by Rietveld method. We provide in this new version a semiquantitative estimation (in Table 3) of minerals as obtained by RIR method. We also added this information in section: 3. Methods.

-        It would be beneficial to determine the chemical composition of the samples or some from similar groups.

We agreed, but due to the small size of the samples it was not possible.

Reviewer 2 Report

The paper presents a study concerning the deterioration effects of bricks masonry in the Venice lagoon cultural heritage. The paper is well-written, and the results of the experiment are well-presented.

Figures and tables are also adequately presented. Overall, the authors demonstrate mastery of the topic This reviewer does not rise critical comments against the publication of this paper.

Some minor comments are listed below:

1) check the email address of the corresponding author.

2) in my opinion the abstract is too long. In my opinion, authors might be more concise.

3)Introduction section is well-referenced.

4) In section 1.2 authors can detail a bit more the masonry typology more common in the Venice panorama. I recommend referencing this paper for the pattern classification https://doi.org/10.1016/j.dibe.2023.100119

Author Response

#reviewer2

The paper presents a study concerning the deterioration effects of bricks masonry in the Venice lagoon cultural heritage. The paper is well-written, and the results of the experiment are well-presented.

Figures and tables are also adequately presented. Overall, the authors demonstrate mastery of the topic. This reviewer does not rise critical comments against the publication of this paper.

Some minor comments are listed below:

1) check the email address of the corresponding author.

Done. Thank you!

2) in my opinion the abstract is too long. In my opinion, authors might be more concise.

According the consideration also of the reviewer1 we deleted some details on the hyperspectral ranges. We think in the abstract there are all the information useful for potential readers, thus we decide not modify it deeply.

3) Introduction section is well-referenced.

Ok. Thank you.

4) In section 1.2 authors can detail a bit more the masonry typology more common in the Venice panorama. I recommend referencing this paper for the pattern classification https://doi.org/10.1016/j.dibe.2023.100119

Thank you for the suggestion. This study mainly focuses on the material characterization, and we are not expert in masonry typology. Anyway, we are now working with some colleague of the Dept. of Engineering of Padova in order to simulate the load stress of the façade according also to our results (and exploiting 3D images already performed).